# From Theoretical Network to Bedside: Translational Application of Brain-Inspired Computing in Clinical Medicine

Tinen L. Iles

Department of Surgery, University of Minnesota, Minneapolis, MN 55455, USA; thealy@umn.edu

**Abstract:** Advances in the brain-inspired computing space are growing at a rapid rate, and many of these emerging strategies are in the field of neuromorphic control, robotics, and sensor development, just to name a few. These innovations are disruptive in their own right and have numerous, multi-dimensional medical applications within precision medicine, telematics, device development, and informed clinical decision making. For this discussion, I will define brain-inspired computing in the scope of simulating the architecture of the brain and discuss the realization of integrating hardware and other technologies with the applications of medicine, along with the considerations for the regulatory pathway for approval and evaluating the risk/consequences of failure modes. This perspective is a call for continued discussion of the development of a pathway for translating these technologies into medical treatment and diagnostic strategies. The aim is to align with global regulatory bodies and ensure that regulation does not limit the capacity of these emerging innovations while ensuring patient safety and clinical efficacy. It is my perspective that it is and will continue to be critical that these technologies are correctly perceived and understood in the lens of multiple disciplines in order to reach their full potential for medical applications.

**Keywords:** computing; engineering; translational science; medicine





## 1. Perspective

Brain-inspired computing in the scope of this text will be defined as simulating the architecture of the brain [1]. These applications for medical use are a critical part of the *future for patient care*; they will be driven by the level of *predictive clinical efficacy* and will require pairing *surrogate measures* that correlate with clinical outcomes.

Features of cognitive intelligence in brain-inspired computing and neuromorphic control can exist as an architecture and be realized in medicine; however, bridging the gap between what is theoretically possible and the bedside requires multiple phases of consideration. I will define a framework for this translational pathway as follows (Figure 1). First, **developing the architecture** and testing the applications, for example, as a Neural Engineering Framework (NEF) [2]. Secondly, this can be **implemented (realized) in hardware** and, thirdly, **further integrated with other technologies**, such as microfluidics [3,4], and embodied with mixed analog and digital signals in end-to-end robotics [5]. The fourth phase within these technologies is **ideation for medical device design and patient care strategies** that range from prediction for intervention to end-to-end adaptive medical devices and prosthetics. This will take the work of multi-disciplinary teams to ideate the applications and needs statements. Lastly, it is essential to discuss the considerations for the **regulatory process and post-approval surveillance**, as it can be foreseen that many of the current global regulatory guidelines may need to be updated because of the highly adaptive capacity of brain-inspired computing. This is an advantage of the technology, but, in many instances, indirect for validating safety and efficacy.

This higher-arching translational pathway is a perspective on how we, as a scientific community, can bridge the gap from theoretical framework to patient care. I will later discuss a more detailed proposed framework that spans multiple disciplines, incorporating

standard medical and engineering processes. This translation in itself is a huge jump, and the proposed framework can benefit our collaborations by understanding what each discipline needs to move forward with innovation in a specific research area.

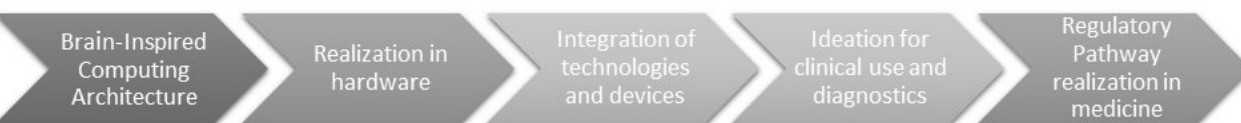

**Figure 1.** Considerations for the pathway of the realization of brain-inspired computing in medicine.

## 2. Brain-Inspired Computing in the Future of Healthcare

There is an endless array of clinical applications when it comes to brain-inspired computing, ranging from disease treatment and implantable devices to diagnostics. The biggest advantage of emerging brain-inspired technologies is the novel capacity for being highly predictive, adaptive, and efficient, and being largely event-based [2,6,7]. There is a lot to unpack in this—the main takeaway is that a clinical platform with these technologies can reach a new dimension of connectivity that allows for multi-disciplinary efforts to synergize for holistic (individual) patient care and humanitarian (global) health efforts that could not be implemented previously.

As these combined technologies are employed to interface with humans, it is important to capture their potential in collaborative efforts to be inclusive in the use of brain-inspired computing—the regulatory bodies, the clinicians, policy makers, and advocates—so that even those who are not experts can leverage these advances. The realization of the architectures in brain-inspired computing within hardware and paired with other devices is essential for translation from the theoretical network to the bedside. This will require continued development from the software architecture, to hardware and integration with medical devices or diagnostic tools. Along with the challenges of this realization, it is my perspective that hurdles will arise in the Verification and Validation (V&V) process. As seen from the engineering perspective, Verification is intended to check design specifications, whereas Validation is intended to ensure that the operational needs of the user are being met. This in itself is a challenge, but it becomes even more dynamic with the integration of architectures with hardware and integrating with a device that may be implantable or diagnostic in nature.

Beyond this brief description of the advantages of these technologies, what are some examples and how can they be implemented? Some of the current innovations that integrate brain-inspired architecture can be combined with microfluidics, sensors, robotics, and deep learning (Table 1); it should be noted that, although there is a primary advancement within one of the fields, it adds vast complexity to development because it integrates multiple advancements in series. As an example with sensors, there has been an immense advancement utilizing Spiking Neural Networks (SNN) and the precision of control with the use of NEF. There are also enormous clinical applications when it comes to the precision of robotics for human–machine interfaces and surgical robotics.

**Table 1.** Examples of implementing brain-inspired architecture with emerging technologies—considerations for verification and validation when translating to medicine.

|  | Microfluidics | Sensors | Robotics | Artificial Intelligence |
|---|---|---|---|---|
| Emerging Technology | Combined programmable platforms | Sensors combined with SNN | Neural Engineering Framework (NEF)-based robotics | Deep learning network architectures, fusion strategies |
| Example Medical Applications | Computer-aided designed diagnostics with hybrid technologies [4,8] | Highly sensitive prosthetics [9] | Advancements in robotics [2,10] | Fusion of CT/MRI in medical imaging [11] |
| Considerations | Validation of sensitivity for specificity for real time | Risk will shift based on Context of Use | Multiple-system validation | Complex validation when attached to outcomes |

## 3. Clinical Implementation: Human–Device Interface, Medical Devices, Clinical Diagnostics, and Clinical Decision Support

Before we begin our discussion of the regulatory applications or technical aspects of Verification and Validation, I will note four main ways that brain-inspired technologies can be employed in medicine: (1) Human–machine interfaces, (2) Medical devices, (3) Clinical diagnostics, and (4) Clinical decision support.

*Human–device interface/wearables:* Input from a device or sensor that interfaces with conscious or unconscious human decision-making and may incorporate embodiment or other Artificial Intelligence (AI), and end-to-end decision making, as demonstrated in robotics applications in prosthetics.

*Implantable medical device:* The term medical device captures the breadth of all of these categories; the difference here between interfacing and implantable medical devices is that the device may be adaptive to physiology, but does not require it; for example. brain-inspired computing in implantable medical devices is an adaptive pacing strategy with the use of SNN. The concept is to benefit the patient with improved long-term outcomes from a more physiological pacing strategy along with more efficient energy use [6].

*Clinical diagnostics:* Creating diagnostic tools with improved specificity and sensitivity, including cancer diagnostic tools, counting cells, aggregates, or testing other biological materials for diagnostic purposes. Combining novel computational frameworks with microfluidics is an example of using diagnostic strategies [3,4,7].

*Informed clinical decision support:* A clinical decision can be supported by the evaluation of the risk of the outcomes that are largely weighted on the strength of a prediction. These data can inform an event (series of events) that is related to the predictive efficacy of a treatment/strategy and/or patient outcome [12,13]. Clinical decision support can be specific to the patient, for example, computational simulation of specific anatomy or derived from clinical databases. The risk/consequence of the decision can be evaluated by the clinical care team, and the weight of the decision from the model is incorporated in risk quantification.

## 4. Perspective on Verification, Validation, and Risk Assessment for Brain-Inspired Computing

Verification and Validation (V&V) processes are independent procedures that are used together for ensuring that a product, service, or system meets requirements and specifications, and that it fulfills its intended purpose [14,15]. It is important to define under what conditions the technology (or combined technologies) will be used—the Context of Use (COU). In medicine, the COU is critical for quantifying safety, efficacy, and risk to the patient. This text is not meant to guide the development of V&V programs, but evaluate how this process may be translated for use specifically in brain-inspired computing strategies.

Industrial companies have an internal V&V process to ensure the safety and efficacy of their product and define failure modes and risk. Many of the guiding technical documents are online, and there are end-to-end examples available [9]. There are many international committees that work together to guide global regulatory standards. A few examples include the American Society of Mechanical Engineering (ASME), which is represented as V&V committees, VV10-70, ranging from solid mechanics to Artificial Intelligence. There has also been recent updated guidance from the Food and Drug Administration (FDA) on the use of Software as a Medical Device (SaMD), in addition to the European Economic Area (EEA) and the International Medical Device Regulators Forum (IMDRF) [16].

## 5. Consideration for Computational Modeling and Simulation Relative to Brain-Inspired Computing Architectures

*Computational Simulation:* Computational modeling can be used as a tool to evaluate multiple physical elements that affect human physiology and anatomy. For example, computational simulation may be used to predict the vorticity or wall shear stress (WSS), which is a predictive surrogate marker for the development of a thrombus. Computational modeling can be used in combination with clinical outcomes and is the subject material

of V&V40: Assessing Credibility of Computational Modeling through Verification and Validation: applications to Medical Devices. These working groups utilize a framework for investigators to establish the credibility of the computational model for a specific COU and build on documented evidence from evaluating the model risk and identifying the credibility goals in order to create a plan for assessing credibility. In the context of brain-inspired computing, there are numerous applications that will overlay this process with the validation of the framework itself For example, a model of cardiac function where there is an in silico trial and/or patient-specific prediction of an adaptive pacing strategy to predict long-term outcomes, or many other paradigms where there is a model in a model. It is important to consider how the models behave and how they reflect reality.

*Software as a Medical Device (SaMD):* There have been many recent updates to the language and definitions of SaMD—the current, non-binding recommendations for SaMD can be found here: Available online: https://www.fda.gov/media/154985/download (accessed on 28 February 2022), which describes SaMD as a medical device intended to be used for one or more medical purposes without being part of the hardware medical device. However, it also needs to be taken into consideration that SaMD can be in SiMD (software in a Medical Device). Within computational modeling and simulation, there is also an outline for in silico trial and Computational Modeling & Simulation (CM&S)-qualified tools for developing or validating a medical device.

*Risk:* There are multiple different conversations regarding "risk." There is a risk associated with consequence, or decision consequence; for example, if a prediction is wrong, what is the risk to the patient? Secondly, how much is the model informing the clinical decision? The risk will be inherently higher if an adaptive, autonomous device is making decisions in real time. However, if the risk to the patient is low if the device fails, then there is an overall lower risk. If a simulation is supporting a decision that is being processed with a novel framework, it will require a body of evidence to reflect the COU. This will require carefully designed clinical and in silico trials to merge multiple strategies for assessing the risk of device failure and risk to compromised predictive efficacy, in addition to modifications in post-market surveillance strategies. An example of a framework to evaluate the significance (weight) of the use of a SaMD and the risk to the patient is outlined in a figure from the International Regulators Forum that highlights the significance of the use of a SaMD relative to the patient's condition and risk associated with the failure of the clinical decision (Table 2). For example, if a patient is in a critical condition and the SaMD is being used to treat or diagnose the patient, the significance (risk/consequence) of the failure of the SaMD could be catastrophic. However, if the SaMD is only part of the picture that is informing the clinical management plan, then there is a lower risk if the SaMD fails.

**Table 2.** International Regulators Forum SaMD Risk Framework (retrieved from International Medical Device Regulator Forum, IMDRF). The left column represents the state of the condition relative to the significance of the information from the SaMD driving medical care. The roman numerals (I–IV) describe the risk (significance) associated with the clinical situation, with I being low and IV being high.

| State of Healthcare Situation or Condition | Significance of Information Provided by Samd to Healthcare Decision | | |
|---|---|---|---|
| | Treat or Diagnose | Drive Clinical Management | Inform Clinical Management |
| Clinical | IV | III | II |
| Serious | III | II | I |
| Non-serious | II | I | I |

## 6. From Theoretical Network to Bedside: An in-Progress Framework for Medical Regulation in Brain-Inspired Computing

In developing a framework for a translational pathway for brain-inspired computing, I believe we must merge existing frameworks [17,18] and complete end-to-end examples

with technologies in the fields of (1) Human–machine interfaces, (2) Medical devices, (3) Clinical diagnostics, and (4) Clinical decision support. This research will elucidate the areas that need to be more closely examined for the safety and efficacy of the devices and developing models of risk/consequence, in addition to fostering ideation in the utility of brain-inspired frameworks in medicine. In Figure 2, I propose a framework for brain-inspired computing in medicine that merges the frameworks from engineering and medical practices to identify surrogate measures when it is not possible to obtain a direct measurement and a systematic approach.

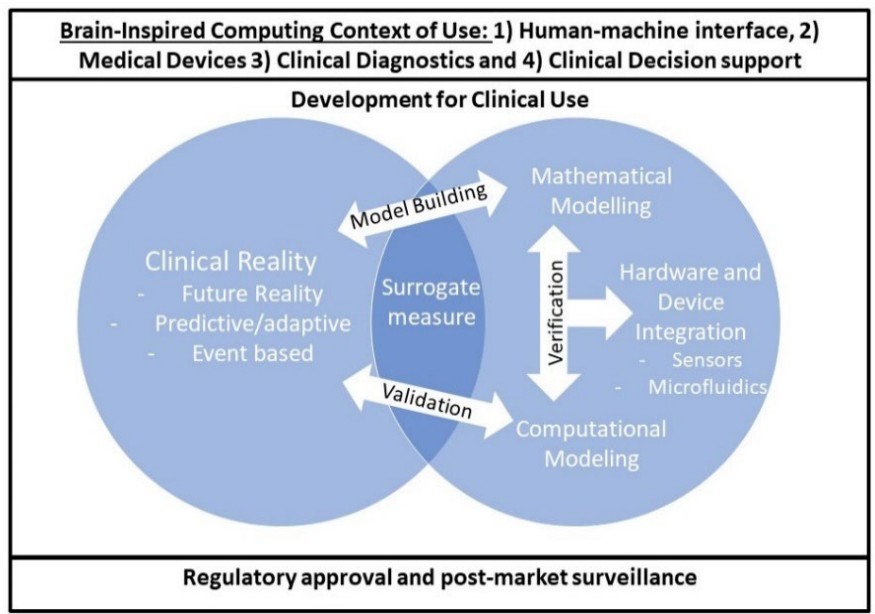

**Figure 2.** Proposed a framework for the ideation and realization of brain-inspired computing in medicine.

## 7. Conclusions

There is a need for the scientific community to consider the complexity of a pathway for translating theoretical frameworks and brain-inspired technologies into medical treatment and diagnostic strategies. I believe there will be a high impact on the adoption of these strategies if we take the time to consider how to align with global regulatory bodies and ensure that regulation does not limit the capacity of these emerging innovations while ensuring patient safety and clinical efficacy. Game-changing ideas and innovations need to bridge the gap. The appreciation of these complexities alone can support the innovation process, in addition to seeking intentional conversations across disciplines in such a way that we can work together to overcome these challenges and combine our frameworks to leverage the essence of the applications of cognitive intelligence in brain-inspired computing.

**Funding:** This research received no external funding.

**Institutional Review Board Statement:** Not applicable.

**Informed Consent Statement:** Not applicable.

**Data Availability Statement:** Not applicable.

**Conflicts of Interest:** The author declares no conflict of interest.

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
