# Peer review of "From Theoretical Network to Bedside: Translational Application of Brain-Inspired Computing in Clinical Medicine"

_applsci, doi:10.3390/app12125788_

Round 1
Reviewer 1 Report
The paper titled "From Network to Bedside: Translational Application of Brain-Inspired Computing
in Clinical Medicine" provide a thorough survery related to the brain-inspired computing in clinical
medicine. However, I have a few doubts.
1. What is "brain-inspired computing"?
I do not get what the author refers with the word "brain-inspired computing". Deep learning can
be one of them, and neural modeling or brain simulation can be. Neural Networks (as the author
mentioned SNN) can be.
Please define its range.
2. The title, why "from network"?
The title "from network to bedside" is somewhat vague. I think "to bedside" is a metaphor for
clincal use in close. However, what is the intention of "from network"? network of what?
Network of neural "networks"? I don't get it.
3. proposed framework. what is good for?
I think it is the author's final proposition. However, due to lacks of explanation, I don't get its
benefits. Especially, what is difference between mathematical modeling and computational
modeling? and why transition between them leads hardware and device integration?
What is the difference between verification and validation? Why is transition between them
verification? There is no mention of the mathematical modeling in the main texts and thus,
there is missing discussion behind it. I feel a hugh logical jump.
Minors
1. Page 2, 3rd paragraph, "Neural Engineering Frameworks (NEF)" -> NEF
(This abbreviation was already introduced in the first page.)
2. Page 3. "Perspective on Verification, Validation and Risk Assessment for Brain-Inspired Com-
puting" -> This should be a heading.
3. Page 5. "From Network to Bedside: an in-progress framework for medical regulation in brain-
inspired computing" -> This should be a heading.
4. Page 4. "In most VV contexts, computational modeling works that numerical modeling informs
the computational model and the computational model is validated."
I don't get the sentence. Please clarify it.
5. Page 4. "Risk", this sub-heading is not parallel to the others "Computational simulation" and
"SaMD". Isn't it the part of SaMD?
6. Page 5. Figure 3. What is this for? There is no explanation about the table in the main texts,
and figure legends. it's hard to understand. What is the meaning of Roman numbers in the
table? What is the definition of state of health care situation: critical, serious, non-serious?
7. Page 5, "In figure 3" -> "In figure 4"
8. Page 6. "Figure 3. Proposed a framework" -> "Figure 4. A proposed framework"
Reviewer 2 Report
This article deals brain-inspired computing in the scope of simulating the architecture of the brain and discuss the realization of integrating hardware and other technologies in the applications of medicine; along with the considerations for the regulatory pathway for approval and evaluating the consequences of failure modes. The aim is to align with global regulatory bodies and ensure that regulation does not limit the capacity of these emerging innovations while ensuring patient safety and clinical efficacy.
However, there are many fundamental aspects that need to be improved.
However, there are many fundamental aspects that need to be improved. In principle, the work in its current state does not show the contribution qualitatively or quantitatively.
1.- I suggest that the literature be analyzed in depth to retrieve useful conclusions of interest to the scientific community.
2.- Use a vocabulary closer to the disciplines addressed in the work to show a deep analysis of the subject.
3.- Improve the writing in the English language to facilitate the reading of the document.
4.- It is also important to analyze and show the important information of the international regulation on Application of BrainInspired Computing in Clinical Medicine.
5.- Show the current technical challenges to implement BrainInspired Computing in Clinical Medicine
Round 2
Reviewer 1 Report
Dr Tinen tried to resolve the issues I raised and improved the manuscript significantly.
The author did not mark all the changes with yellow highlights, and thus I reviewed the
whole manuscript thoroughly again.
I realized that I am the sole reviewer of this paper, and fast publishing is one of the
virtue of MDPI journals, I feel that it is not a good idea to hold this article any longer.
However, I think it still requires some changes.
1. Brain-inspired computing...
Even though it's a bit hard to agree with the definition of Dr Tinen's view, since it is
her idea and the baseline of this paper, I would not raise issue any more.
Just provide a reference to the assertion, please.
2. title,
I really appreciate the consideration. "From theoretical framework to bedside" is at least
acceptable to me. However, on the main text, the corrected title (yellow highlighted title)
is "from theoretical network to bedside". Which one is the author's final call?
Also, since the title is updated, the similar mentions in the main text should be updated, too.
Page 2. second paragraph, line 7. "From the theoretical network to bedside" -> "from theo-
retical framework to bedside"
Page 5. Heading 6. "From Network to Bedside" -> "From theoretical framework to Bedside"
3. Proposed framework.
Understandable now.
minor
page 2. second paragraph, line 7. willrequire -> will require
Page 3. Paragraph of "Human-device interface",
Input from a device,sensor, that interfaces -> Input from a device (sensor) that interfaces
(AI), end-to-end -> (AI), end-to-end (double blanks)
Page 3. Paragraph of "Implantable Medical Device", missing "." at the end of the paragraph.
page 3. Paragraph of "Clinical Diagnostics",
frameworkswith -> frameworks with
I think it's better to revise in order to widen the concept of clinical diagnostics, and keep
parallelism.
"Creating diagnostic tools with improved specificity and sensitivity "including" diagnosing
cancer, counting cells, and testing with other biological materials for diagnostic purposes. "
Page 3. Paragraph of "Informed Clinical Decision Support".
This data can -> This data can (double blanks)
page 5, last paragraph, line 8. frameworksfrom -> frameworks from
There are so many punctual errors beside above. Please check thoroughly.
